# Management of Anaemia of Chronic Disease: Beyond Iron-Only Supplementation

**DOI:** 10.3390/nu13010237

**Published:** 2021-01-15

**Authors:** Evasio Pasini, Giovanni Corsetti, Claudia Romano, Roberto Aquilani, Tiziano Scarabelli, Carol Chen-Scarabelli, Francesco S. Dioguardi

**Affiliations:** 1Cardiac Rehabilitation Division, Scientific Clinical Institutes Maugeri, IRCCS Lumezzane, Lumezzane, 25065 Brescia, Italy; evpasini@gmail.com; 2Division of Human Anatomy and Physiopathology, Department of Clinical and Experimental Sciences, University of Brescia, 25065 Brescia, Italy; cla300482@gmail.com; 3Department of Biology and Biotechnology, University of Pavia, 27100 Pavia, Italy; dottore.aquilani@gmail.com; 4Center for Heart and Vessel Preclinical Studies, St. John Hospital and Medical Center, Wayne State University, Detroit, MI 48202, USA; tscarabelli@hotmail.com; 5Division of Cardiology, Richmond Veterans Affairs Medical Center (VAMC), Richmond, VA 23249, USA; chenscarabelli@hotmail.com; 6Department of Internal Medicine, University of Cagliari, 9128 Cagliari, Italy; fsdioguardi@gmail.com

**Keywords:** chronic diseases, iron deficiency, haemoglobin, anaemia, aminoacids, rehabilitation

## Abstract

Chronic diseases are characterised by altered autophagy and protein metabolism disarrangement, resulting in sarcopenia, hypoalbuminemia and hypo-haemoglobinaemia. Hypo-haemoglobinaemia is linked to a worse prognosis independent of the target organ affected by the disease. Currently, the cornerstone of the therapy of anaemia is iron supplementation, with or without erythropoietin for the stimulation of haematopoiesis. However, treatment strategies should incorporate the promotion of the synthesis of heme, the principal constituent of haemoglobin (Hb) and of many other fundamental enzymes for human metabolism. Heme synthesis is controlled by a complex biochemical pathway. The limiting step of heme synthesis is D-amino-levulinic acid (D-ALA), whose availability and synthesis require glycine and succinil-coenzyme A (CoA) as precursor substrates. Consequently, the treatment of anaemia should not be based only on the sufficiency of iron but, also, on the availability of all precursor molecules fundamental for heme synthesis. Therefore, an adequate clinical therapeutic strategy should integrate a standard iron infusion and a supply of essential amino acids and vitamins involved in heme synthesis. We reported preliminary data in a select population of aged anaemic patients affected by congestive heart failure (CHF) and catabolic disarrangement, who, in addition to the standard iron therapy, were treated by reinforced therapeutic schedules also providing essential animo acids (AAs) and vitamins involved in the maintenance of heme. Notably, such individualised therapy resulted in a significantly faster increase in the blood concentration of haemoglobin after 30 days of treatment when compared to the nonsupplemented standard iron therapy.

## 1. Introduction

Noncommunicable diseases account for 38 million deaths per year, according to the World Health Organization [1]. Of these deaths, chronic diseases (CD) constitute a major cause of mortality. The most common CD include congestive heart failure (CHF), senescence, cancer, chronic obstructive pulmonary disease (COPD), diabetes, arthritis, asthma and some viral diseases such as hepatitis C and acquired immunodeficiency syndrome [2].

All CD are characterised by a hypercatabolic syndrome due to low-grade inflammation (caused by specific molecules such as cytokine, hormones, etc.), which induces metabolic alterations and muscular and globular protein disarray. An unmatched autophagy activity ensues, clinically resulting in sarcopenia, hypoalbuminemia and hypo-haemoglobinemia (otherwise known as anaemia) [3,4]. Among the globular proteins, haemoglobin (Hb) is one of the most readily measureable in the blood. 

Mounting experimental and clinical evidence has demonstrated that both anaemia and iron deficiency (ID) are present in patients with CD, resulting in the significant limitation of therapeutic rehabilitative strategies such as rehabilitation programs, thereby worsening the prognosis of these patients [5,6,7]. Studies suggest concomitant roles for inflammation and autophagy bridged to the subsequent protein disarrangement [8]. 

Gut dysbiosis, nutritional imbalance (malnutrition) with dysgeusia and, most importantly, ID with or without renal dysfunction capable of reduced erythropoietin-mediated erythropoiesis, are responsible for anaemia in CHF patients [5,9]. Consequently, iron supplementation, with or without the addition of erythropoietin, is the most commonly recommended therapeutical approach to CHF-mediated anaemia [10].

Based on the current biochemical knowledge, the pathogenesis of anaemia in CD (including CHF) should be considered in its entirety [11]. Although heme is the principal biochemical constituent of haemoglobin, with ID contributing partially to the anaemia in CD, one of the additional contributory factors in anaemia includes the impaired synthesis of the tetrapyrrolic rings to which iron binds, thereby facilitating the metabolic function of heme, a hemoprotein. The production of D-amino-levulinic acid (D-ALA), the limiting step in the synthesis of the heme ring, warrants significant consideration. D-amino-levulinic acid (D-ALA) is derived from one amino acid, glycine, and Kreb’s cycle intermediate succinyl-coenzyme A (CoA) [12]. Therefore, an adequate treatment of anaemia in CD necessitates the incorporation of a standard iron infusion, along with the supplementation of essential amino acids (EAAs) and vitamins involved as precursors or cofactors in heme synthesis. 

In order to demonstrate the importance of the molecules involved in the synthesis of haemoglobin, we first provide a review of the main events necessary for the synthesis and the functions of heme and the roles of the molecules centrally involved in heme synthesis. Following the review, we present a clinical study designed to evaluate the effects of an integrated supplementation with iron and the molecules involved in the synthesis of haemoglobin in anemic patients with evident deficiencies of these molecules.

### 1.1. Iron and Heme

Iron (Fe), a requisite metal in almost all biological systems, is necessary for numerous critical processes, such as DNA synthesis, heme and iron-sulfur cluster synthesis, etc. Therefore, the cellular regulation of the iron concentration is essential for the maintenance of normal physiology [13]. 

About 70% of the body’s iron is found in the red blood cells as a component of haemoglobin and in the muscle cells as myoglobin. Iron is also a crucial component of a very large class of metalloproteins containing heme—hence the name, hemoproteins. 

Heme is an organic, ring-shaped molecule consisting of an iron ion coordinated to four pyrroles, which are small, pentagon-shaped molecules with four carbons and one nitrogen, which, together, form an iron-binding tetrapyrrole, called a porphyrin (Figure 1). Thus, heme is an iron-binding porphyrin [11]. Interestingly, iron plays a balanced attractive force interacting with the nitrogen molecules of heme; thus, the electrons stay balanced, and the global molecule remains stable.

There are four different forms of heme in nature: heme-A, -B, -C and -O; they influence the functions of the molecules in which heme is present. Although heme-B is the most common form, heme-A and -C are present in many molecules. The biochemical behaviors of the most common heme groups are regulated by differences of the functional groups in the side chains bound to carbons 3, 8 and 18 [11]. 

### 1.2. The Synthesis of Heme

Porphyrin synthesis, the biochemical pathway from which heme is derived, begins with the synthesis of D-amino-levulinic acid (D-ALA), which is also the limiting step in heme synthesis [12].

D-ALA originates from the amino acid (AA) glycine and from the Kreb’s cycle intermediate succinyl-CoA, which, in turn, is derived by α-ketoglutarate or from the metabolism of the EAAs isoleucine, methionine, threonine or valine. Interestingly, D-ALA synthesis occurs inside the mitochondria and depends on the activity of an enzyme named ALA-synthase, which is negatively regulated by glucose and the heme concentration. Importantly, inhibition of the enzyme is also dependent on the stability and availability of its mRNA in the mitochondria. Notably, AAs are the sole sources of carbon and nitrogen atoms provided to D-ALA, demonstrating the narrow link between the metabolism of the AAs and the energetic metabolism.

Released from the mitochondria, two D-ALA molecules are condensed to form porphobilinogen in the cytoplasm. This synthetic reaction continues until the formation of coproporphyrinogen-III, which is transported back inside the mitochondrial matrix and converted into protoporphyrin-IX. The enzyme ferrochelatase then inserts an iron atom, forming one heme molecule, which is then shuttled to the cytoplasm, where it is available for the synthesis of heme-based molecules [14]. The process of heme synthesis is illustrated in Figure 2.

### 1.3. Functions of Heme

Heme and hemoproteins have many biological functions. The presence of an iron atom serves as a source of electrons during electron transfer or redox chemistry, thereby giving heme the ability to transport biatomic gases and to exert a chemical catalysis requiring an electron transfer.

Hemoproteins participate in many diverse biological actions (such as oxygen transport) fundamental for life. Indeed, although haemoglobin and myoglobin are the two best-known hemoproteins, other important, although often overlooked, enzymes that belong to hemoproteins include cytochrome p450s, cytochrome-c oxidase, cyclooxygenase 2, catalase, peroxidases and endothelial nitric oxide synthase. In addition, as part of the electron transport chain, hemoproteins also enable an electron transfer. A change in the iron content affects important cell survival systems, illustrating that heme is not only pivotal for oxygen transport but also plays a fundamental role in other important metabolic pathways, such as energy production, the transformation of many molecules, the detoxification of aggressive molecules such as as oxygen-free radicals, the regulation of inflammation and/or vascular tone and blood coagulation [11].

### 1.4. Other Molecules Involved in Heme Synthesis 

CD, especially if associated with qualitative malnutrition, induces a hypercatabolic state and consequent protein disarrangement, which can precipitate the development of anaemia secondary to a reduction in haemoglobin. A schematic representation of this link is proposed in Figure 3. Independently from iron, other molecules that are strictly related to heme synthesis include:Vitamin B1. Its pyrophosphate ester, thiamine diphosphate (TPP), is a cofactor for enzymes that catalyse alpha-keto acids of molecules involved in the Kreb’s cycle and its intermediary metabolism [15].Vitamin B6. It cocatalyses reactions related to the anabolism and catabolism of AAs, facilitating the reactions of transamination. Interestingly, it is involved in protein folding and interacting with the folate cycle. In addition, vitamin B6 is a scavenger of free oxygen radicals [16].Vitamin B9 (Folate). It is a cofactor of many enzymes involved in the redox reactions and transfer of the AA one-carbon unit (DNA methylation) [17].Vitamin D. It has anti-inflammatory properties, reducing circulating cytokines (interleukin 6 (IL-6) and IL-1B) that counteract catabolism and autophagy. It stimulates the synthesis of anabolic molecules (such as fibroblast growth factor-23 (FGF-23)) and increases red blood cell lifespans. In addition, it modulates hepcidin, a molecule responsible for the regulation of iron metabolism [18].

Amino Acids. Hemoproteins (as haemoglobin), consisting of heme (as the metabolically active part) and the surrounding proteins (as globin molecules), contain a large number of different AAs. Previous studies have demostrated that the administration of mixtures of free EAAs tailored to match the human metabolic process were able to improve anabolism, the aerobic metabolism and mitochondrial neogenesis [19,20]. Rapidly absorbed, this mixture contains appropriate stoichiometric amounts of all EAAs that can be converted into nonessential AAs (NEAAs), such as glycine [21]. Moreover, the EAAs mixture contains: (a) L-leucine, which modulates the enzyme mTORC1 involved in haemoglobin production, and (b) histidine, which stabilises bound O_2_ and, placed in particular positions, acts as a gate, allowing ligands entry into both haemoglobin subunits [22,23]. Therefore, histidine, essential in globin synthesis and erythropoiesis, has also been implicated in the enhancement of iron absorption from human diets. Furthermore, histidine has already been proven to be effective both in improving the antianaemic efficiency and limiting the damages resulting from an iron overload and oxidative stress caused in chronic kidney disease (CKD) [24]. Conversely, beta-alanine supplementation would impair protein synthesis by reducing the histidine concentration and availability [25]. 

In light of these considerations, the provision of the molecules involved in the synthesis of heme and haemoglobin is essential, even more so if patients are malnourished.

## 2. Methods 

Based on the aforementioned fundamental biochemical knowledge, and in observation of “good medical practice” (www.gmc-uk.org), we conducted a controlled clinical trial that integrated personalised standard therapy with an iron infusion, along with the administration of specialised mixtures rich in free EAAs [21] and vitamins (B1, B6, B9 and D) to treat heme synthesis deficiency in a cohort of select elderly female patients (*n* = 15; ages 78.3 ± 8.5 years old (y.o.)) with signs and/or symptoms of CHF with a preserved ejection fraction (HFpEF). Written informed consent was obtained; ethical approval was not required under local legislation. The inclusion criteria were: (1) anaemia (Hb > 8.5/<11.5 mg/dL); (2) symptoms and signs of stable CHF for at least 3 months on a standard medical therapy with a beta-blocker, diuretics, ACE-inhibitor or ARB (Angiotensin Receptors Blockers); (3) protein disarrangement (serum abumin < 3.5 g/dL) but a normal body mass index (BMI) (range >23 <30 for people over 65 y.o.); (4) iron deficiency (plasma iron < 50 µg/dL, ferritin < 100 ng/mL or serum ferritin within the range 100–299 μg/dL when the transferrin saturation is < 20%); (5) inflammation (by C-reactive protein (CRP) > 5 mg/dL) and (6) vitamin D and/or folate lower than the normal ranges of the serum concentrations (15.2–90.1 pg/mL and >3.00 ng/mL, respectively). 

In addition to the low haemoglobin concentration, since these patients had serum albumin and vitamin D and/or folate levels below the minimum, they were treated, according to good medical practice standards, for 30 days with a daily intravenous administration of 2 mL of ferric carboxymaltose containing 100 mg of elemental iron (50 mg/mL), integrated with an oral administration of 4g of a specific free AAs mixture rich in essential ones (84%) containing 0.15 mg of vitamin B1 and 0.15 mg of vitamin B6, 15 mg of vitamin B9 (as a calcium folinate tablet) and 1000 IU (25 µg)/day of cholecalciferol (vitamin D). The cumulative weekly dose of elemental iron administered was 700 mg (less than the maximum dose indicated by the manufacturer, which corresponded to 1000 mg/week).

Intravenous ferric carboxymaltose previously demonstrated improved symptoms, functional capacity and quality of life in HF patients, even in the absence of anaemia, in the FAIR-HF clinical trial [10]. Subsequent clinical trials re-confirmed the benefits of intravenous ferric carboxymaltose, with improvements in exercise capacity in the EFFECT-HF trial [26] and a reduced risk of hospitalizations for HF exacerbation in the CONFIRM-HF trial [27] and in the recently published AFFIRM-AHF clinical trial [28].

The control group consisted of a cohort of elderly female patients (*n* = 15; age 76.1 ± 11 y.o.) with the same inclusion criteria, except that the levels of albumin and vitamins were near the lower limits but still within the normal range. This group received only standard iron therapy without supplementation. The baseline mean clinical biochemical data from two cohorts are summarised in Table 1.

### Statistics

Data are expressed as the mean ± standard deviation. A two-tailed paired Student’s *t*-test was used to compare the data before (T0) and after (T30) the therapy within each group. Increments by treatments between the two groups were evaluated by the unpaired two-tailed Welch’s *t*-test. The *p*-value < 0.05 was considered significant.

## 3. Results

The baseline and postintervention clinical biochemical data of the patients reciving integrated therapy are summarised in Table 2. Increased levels of sideraemia (serum iron), ferritin and saturated transferrin were observed in both groups (Table 2 and Table 3). However, when compared to the basal levels, only the experimental group receiving intravenous iron therapy plus integrated therapy demonstrated a significant increase in the haemoglobin concentration (Figure 4A,B), while no modification was observed in the group receiving the standard iron therapy only (see Table 3 and Figure 4C,D). The comparisons of the modifications of the biochemical data for the two groups of patients (postvalue minus prevalue) are summarised in Table 4.

## 4. Discussion and Conclusions

The incidence of anaemia (32%) is common in HF patients, with concurrent iron and folate deficiencies reported in 43% of anaemic patients and in 15% of nonanaemic patients [29]. Currently, the standard therapy of anaemia is primarily based on the supplementation of iron, with or without erythropoietin for hematopoiesis stimulation. Previous randomised, controlled studies with intravenous iron in HF patients reported that haemoglobin increased after four or six months of treatment [7,10,30,31]. Indeed, if a deficiency of fundamental molecules (such as amino acids and vitamins) results in the lack of heme synthesis, iron supplementation alone will not lead to a proportional hemoprotein increase and of Hb in primis. In addition, an isolated increase in soluble iron ions, without any accompanying augmentation in heme, could favor the persistence of oxidative stress (via Fenton/Haber-Weiss reactions), chronic inflammation and autophagy [32]. The mode of iron supplementation also appears to be important, as oral supplementation was ineffective in improving the exercise capacity in HF patients with reduced ejection fraction and iron deficiency [33].

After the integrated therapy, we observed improvements of ferritin in both groups, but this was more elevated in patients who received the standard therapy. Iron, although it is an essential micronutrient, is potentially toxic for biological systems, since it generates free radicals by interconversions between ferrous (Fe^2+^) and ferric (Fe^3+^) forms. However, existing regulatory processes in the body are efficacious in reducing the toxicity of iron, even in response to an overload [34]. Interestingly, patients receiving the integrated therapy had lower ferritin concentrations than those receiving the standard therapy. This suggests that the use of iron for the synthesis of haemoglobin was rapidly promoted and much more effective in the integrated therapy, thus also limiting the possible toxic effects of iron deposits.

Based on these preliminary data showing a rapid escalation in the haemoglobin level (within 30 days after the interventions aimed at increasing both iron and heme), we propose that a more effective approach to treating heme synthesis (including anaemia) in CD must consider not only the iron availability but, also, integrate a therapeutic strategy that counteracts the catabolic drives and promotes protein syntheses. Therefore, the standard intravenous or oral iron supplementation should incorporate also a supply of specific mixtures of EAAs and vitamins involved in the biochemical pathway of heme synthesis, as illustrated in Figure 5. Consequently, the careful evaluation of either the nutritional status of patients and the presence of catabolism or impairement of the availability of molecules involved in heme synthesis, as well as their integration, must therefore be the first step of an effective individualised therapeutic intervention aimed at correcting the state of anaemia in patients with CD such as CHF. Our therapeutical approach based on biochemical data should be confirmed in a large-scale clinical trial.

The main messages:In chronic hypercatabolic diseases (such as CHF), the metabolism of iron and the heme protein is markedly impaired, inducing anaemia and the likely impairment of many other hemoproteins involved in the essential metabolic pathways.Heme is the metabolically active part of haemoglobin. It is characterised by the presence of iron atoms linked to tetrapyrrole groups.Many other important biologically active molecules, named hemoproteins (including Hb), contain heme as the metabolically active part, with the surrounding proteins (as globin molecules) containing a large number of different amino acids.The maintenance of an adequate blood concentration of both iron and heme is fundamental for proper function of the heme-containing enzymes.In patients with anaemia and chronic hypercatabolic diseases, the correction of deficiencies in iron, as well as the metabolic substrates required for all hemoproteins, is essential for proper treatment.

## Figures and Tables

**Figure 1 nutrients-13-00237-f001:**
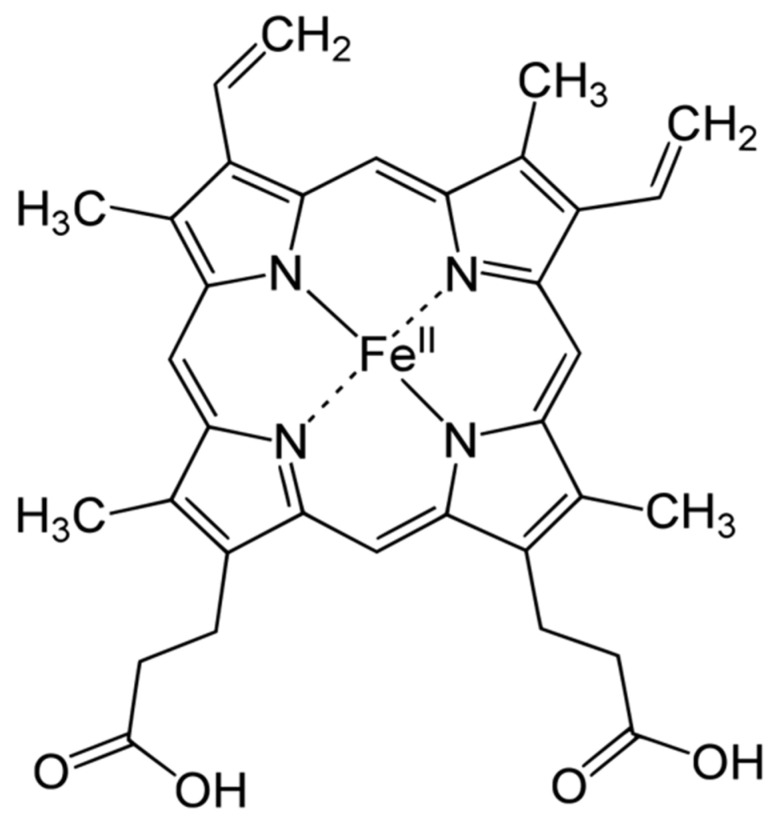
Haemoglobin structure.

**Figure 2 nutrients-13-00237-f002:**
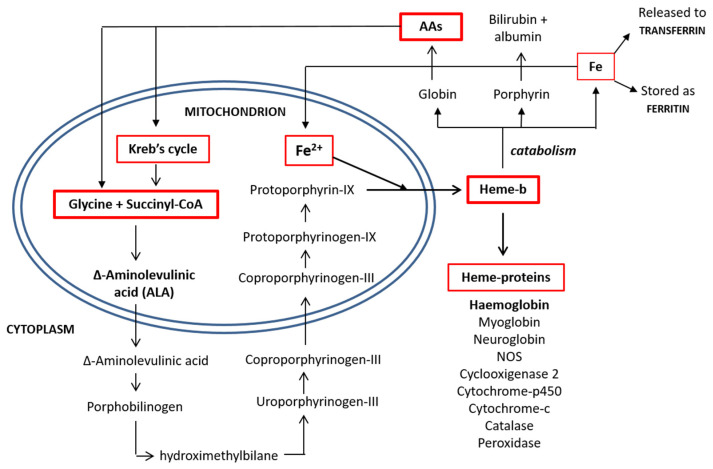
Heme synthesis and degradation pathways. AAs: amino acids and CoA: coenzyme A.

**Figure 3 nutrients-13-00237-f003:**
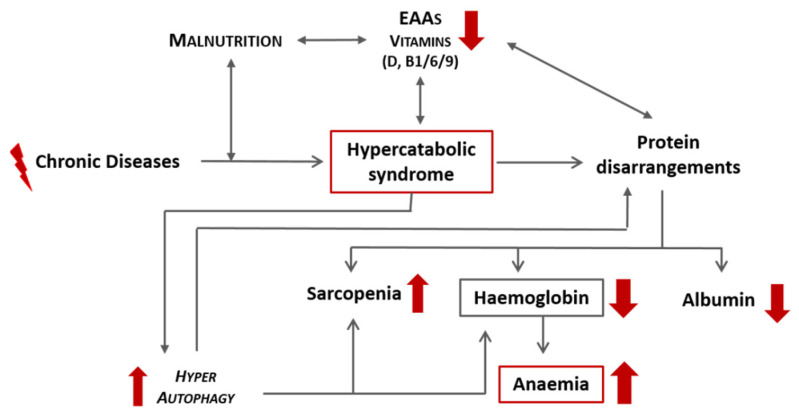
Schematic representation of the effects of chronic diseases and malnutrition on the onset of anaemia. EAAs: essential amino acids.

**Figure 4 nutrients-13-00237-f004:**
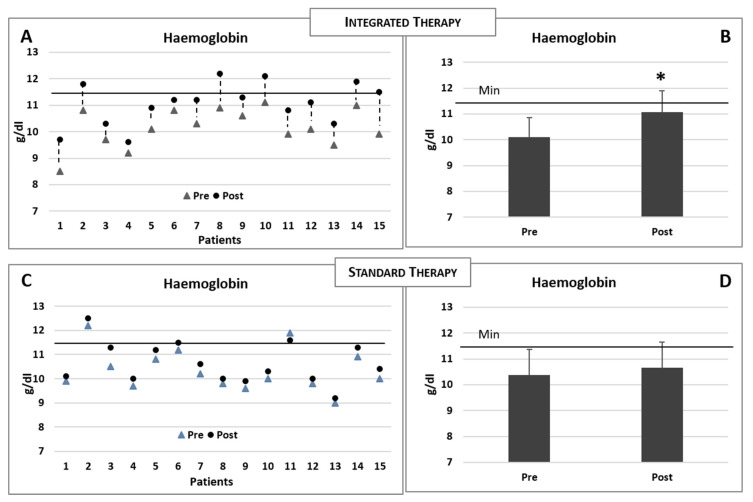
(**A**,**B**) Changes in the haemoglobin concentrations in each patient consequent to the integrated therapy. Histogram shows the mean (±sd) concentrations of haemoglobin before (pre) and after (post) the integrated therapy. (**C**,**D**) Changes in the haemoglobin concentrations in each patient consequent to the iron standard therapy. Histogram shows the mean (±sd) concentrations of haemoglobin before and after the iron standard therapy. The black line indicates the minimum reference value. Two-tailed paired Student’s *t*-test. * *p* < 0.05.

**Figure 5 nutrients-13-00237-f005:**
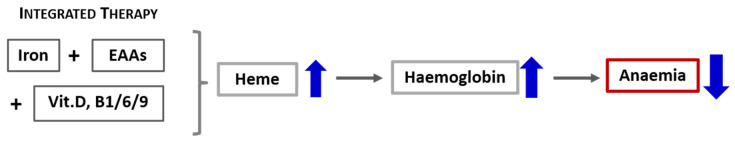
Schematic representation of the effects of integrated therapy on the containment of anaemia.

**Table 1 nutrients-13-00237-t001:** Comparison of the baseline antropometric and clinical biochemical data from patients who received integrated therapy and iron standard therapy (control). Note that patients that received the iron standard therapy had nutritional parameters (albumin and vitamins) close to the lower limit of the normal range. TIBC, Total Iron Binding Capacity; CRP, C-reactive protein; NT-proBNP, B-type natriuretic peptide; LVEF, Left Ventricular Ejection Fraction; BMI, body mass index and CVP, Central venous pressure.

	Integrated Therapy (*n* = 15)	Standard Therapy (*n* = 15)	Normal Value
Age (y.o)	78.3 ± 8.5	76.1 ± 11	---
BMI	26.9 ± 1.85	25.5 ± 1.91	<25
Haemoglobin (g/dL)	10.2 ± 0.8	10.37 ± 0.91	>11.5
Creatinine (mg/dL)	1.06 ± 0.25	1.02 ± 0.19	0,5–1,1
Albumin (g/dL)	3.31 ± 0.37	3.55 ± 0.08	>3.5
Ferritin (ng/mL)	73.73 ± 38.81	94.80 ± 40.76	15–200
Sideraemia (serum iron) (µg/dL)	36.07 ± 6.85	33.72 ± 9.33	50–150
Transferrin saturated (%)	14.33 ± 4.24	12.88 ± 2.18	20–45
Transferrin total (TIBC) (µg/dL)	228.53 ± 48.09	224.33 ± 50.0	180–380
Vitamin B9 (ng/mL)	2.32 ± 0.35	3.13 ± 0.20	3
1,25-OH Vitamin D (pg/mL)	17.73 ± 4.23	21.87 ± 2.25	21–100
CRP (mg/dL)	10.67 ± 2.43	10.82 ± 2.70	<5
NT-proBNP (pg/mL)	2449.2 ± 2048.69	2650.27 ± 2177.65	<450
LVEF (%)	54 ± 6	56 ± 5	>50
CVP (mmHg)	5.5 ± 1.08	5.73 ± 1.05	<8

**Table 2 nutrients-13-00237-t002:** Clinical biochemical data before (baseline) and after (30 days) integrated therapy. TIBC, Total Iron Binding Capacity. Two-tailed paired Student’s *t*-test. * *p* < 0.05.

Integrated Therapy	Baseline	30 Days	*t*	*p*
Haemoglobin (g/dL)	10.1 ± 0.76	11.06 ± 0.83 *	10.94	0.000
Ferritin (ng/mL)	73.73 ± 38.81	390.93 ± 196.4 *	6.89	0.000
Sideraemia—serum iron (µg/dL)	36.07 ± 6.85	81.93 ± 16.84 *	7.59	0.000
Transferrin saturated (%)	14.33 ± 4.24	51.28 ± 10.24 *	13.35	0.000
Transferrin total (TIBC) (µg/dL)	228.53 ± 48.09	253.0 ± 50.37 *	6.14	0.000

**Table 3 nutrients-13-00237-t003:** Clinical biochemical data before (baseline) and after (30 days) the standard iron therapy. TIBC, Total Iron Binding Capacity. Two-tailed paired Student’s *t*-test. * *p* < 0.05.

Standard Therapy	Baseline	30 Days	*t*	*p*
Haemoglobin (g/dL)	10.37 ± 0.88	10.66 ± 0.87	0.834	0.412
Ferritin (ng/mL)	94.80 ± 39.29	631.47 ± 280 *	7.99	0.000
Sideraemia—serum iron (µg/dL)	33.72 ± 9.07	109.0 ± 27.62 *	10.242	0.000
Transferrin saturated (%)	12.88 ± 2.1	56.23 ± 9.5 *	20.368	0.000
Transferrin total (TIBC) (µg/dL)	224.33 ± 48.18	218.33 ± 58.35	1.828	0.089

**Table 4 nutrients-13-00237-t004:** Comparisons of the biochemical data (postvalue minus prevalue for each patient) between the standard therapy and integrated therapy (mean ± st. dev.). TIBC, Total Iron Binding Capacity. Unpaired two-tailed Welch’s *t*-test. * *p* < 0.05.

	Standard Therapy	Integrated Therapy	*t*	*p*
Haemoglobin	0.29 ± 0.22	0.90 ± 0.32 *	6.08	0.000
Ferritin	536.67 ± 260.06	317.20 ± 178.16 *	2.696	0.012
Sideraemia—serum iron	75.28 ± 28.46	45.87 ± 23.39 *	3.092	0.005
Transferrin saturated	45.35 ± 8.24	36.95 ± 10.71 *	2.408	0.023
Transferrin total (TIBC)	13.2 ± 27.97	24.47 ± 15.44	1.366	0.186

## Data Availability

The data presented in this study are available on request from the corresponding author. The data are not publicly available due to privacy reason.

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
