# Peer review of "Management of Anaemia of Chronic Disease: Beyond Iron-Only Supplementation"

_nutrients, 2021, doi:10.3390/nu13010237_

Round 1
Reviewer 1 Report
The manuscript reported the effect of integrated supplementation of iron with amino acids and vitamins on hemoglobin. The findings are potentially meaningful, although there are several shortcomings including statistics.
Major points
1. The statement of conclusion is quite strong, while the statistical method is weak. Subtract the pre-value from the post-value for each subject and compare the standard therapy and the integrated therapy by unpaired Student’s t test or Welch's t test. The outcome will tell the direct evidence of superiority of the integrated therapy.
Line 192-193: Statistics must be placed at the end of the method. The authors should describe both unpaired and paired Student’s t test. Show whether one-sided or two-sided probability was calculated. Show that pre and post value comparison in each group was evaluated by paired t test. Show that increments by treatments between the two groups was evaluated by unpaired t test. If homogeneity of variance will be violated by Bartlett’s test or other correspondence, Welch’s t test should be used instead of Student’s t test. These statistical improvement will reinforce the strength of findings.
2. There are several unreasonable units and parameter names. Some of them might be old-fashioned or unfamiliar names. The amount of vitamins given to subjects is not always reasonable. Each of them is a small deal. They might be careless mistakes. However, they are too many. These facts are encroaching the credibility of this study.
2a. Line 164: unit for iron should be microg/dL
Unit for serum ferritin should be ng/mL or microg/L
Unit for CRP should be mg/dL
2b. Line 166: What is the parameter for vitamin D? It must be shown. What is the parameter for folate? It must be shown.
2c. Describe the amount of elemental iron given to the subjects as well as total amount. If 100 mg is the elemental iron, describe it in the manuscript.
2d. Line 170: Did the authors really give 15mg of Vit.B9? It is really too much considering the requirement.
Did the authors really give 0.15mg of Vit.B1 and Vit.B6? They are quite small compared to the requirements.
2e. Table 1
The % is shown as a unit for Iron. It does not make sense.
The unit for Ferritin should be ng/mL.
What is sideremia? Assuming sideremia is serum iron, a unit of ug/dL (probably microg/dL) does not make sense.
What is Transferrin total? Does it mean Total Iron Binding Capacity? Assuming it is so, mg/dL does not make sense.
What is RCP? Is it CRP? Spell out RCP.
Spell out NT-proBNP in the footnote.
Spell out LVFE in the footnote.
Spell out CVP in the footnote.
2f. Table 2 and Table 3: The same problems in the name of parameters and units exist in these tables as same as Table 1.
3. Introduction is too long. The reviewer gets confused as if this manuscript is for the review paper. Make the description of established findings concise and the hypothesis of this study clear.
4. Discuss the potential hazardous effect of iron excess, because the serum ferritin after the therapy is quite high. In the future, the amount of iron might be reduced without loss of efficacy of the therapy.
5. British spelling (such as haemoglobin) and American spelling (such as anemia) are mixing. Grammatical support is necessary to obtain the integrity of the manuscript.
Minor points
They are mostly errors in spelling. They are shown as error ------ > correct.
Line 21: couses --- > causes
Independentely ----- > ,independent
Line 33: serum concentration ----- > blood concentration
Line 61 tetrapyrrolic proteic rings ----- > tetrapyrrolic rings
Line 142: Lucine ----- > Leucine
Line 171: die ----- > diet
Line 192: Student t-Test ---- > Student t-test
Reviewer 2 Report
Comments:
- the main objective need to be clarified
- the description of "normal BMI>24" need to be clarified. Normal BMI is different for older people. If the authors want to use a range of BMI they need to clarify the range of age.
- The BMI do not represent the evaluation of nutritional status. The authors need to be carefull when made this conclusion.
Round 2
Reviewer 1 Report
The manuscript is much improved. Separation of the review part and the clinical trial part is clarified. British and American spellings are still mixed. British spellings took a major part. The manuscript must be checked by an editing service with British English. The followings are the minor points that must be properly corrected for the publication.
Line 173: albuminemia should be "serum albumin".
Line 176: Insert "of serum concentrations" after "normal ranges".
Line 177: Insert "serum" before "albumin".
Line 179: As mentioned in the first review, state the amount of elemental iron for 100 mg of ferric carboxymaltose clearly.
Line 182: This is the same comment for Line 179. If the amount of 700 mg is as elemental iron, just insert "elemental" before iron.
Line 207: sideremia (iron serum) should be sideraemia (serum iron).
Table 2, 3, and 4: sideremia should be sideraemia.
Table 4: t value for haemoglobin should be 6.084 or 6.083. It depends on the precise values. Anyway, 0.608 is wrong. Please check the data.
Author Response
We thak the referee for his time and attention to the review.
We checked the British English spelling. We also made the suggested corrections by highlighting them in yellow.